# Antimicrobial Resistance Risk Assessment of *Vibrio parahaemolyticus* Isolated from Farmed Green Mussels in Singapore

**DOI:** 10.3390/microorganisms11061498

**Published:** 2023-06-05

**Authors:** Hong Ming Glendon Ong, Yang Zhong, Chengcheng Hu, Kar Hui Ong, Wei Ching Khor, Joergen Schlundt, Kyaw Thu Aung

**Affiliations:** 1School of Chemistry, Chemical Engineering and Biotechnology, Nanyang Technological University, Block N1.2, B3-15, 62 Nanyang Drive, Singapore 637459, Singapore; glendonohm@hotmail.co.uk; 2National Centre for Food Science, Singapore Food Agency, 7 International Business Park, Techquest, Singapore 609919, Singapore; karhuiong7@gmail.com (K.H.O.); khor_wei_ching@sfa.gov.sg (W.C.K.); 3Department of Clinical Translational Research, Singapore General Hospital, Academia, 20 College Road, Singapore 169856, Singapore; yzhong005@e.ntu.edu.sg; 4Singapore Institute of Manufacturing Technology, 08-04, Innovis, 2 Fusionopolis Way, Singapore 138634, Singapore; chengchenghu01@gmail.com; 5Schlundt Consult, 3250 Gilleleje, Denmark; joergenschlundt@gmail.com; 6School of Biological Sciences, Nanyang Technological University, Singapore 637551, Singapore

**Keywords:** antimicrobial resistance risk assessment, emolytic, @Risk, farm-to-home, retail-to-home, sensitivity analysis

## Abstract

*Vibrio parahaemolyticus*, commonly found in seafood products, is responsible for gastroenteritis resulting from the consumption of undercooked seafood. Hence, there is a need to characterize and quantify the risk involved from this pathogen. However, there has been no study reporting the quantification of hemolytic antimicrobial-resistant (AMR) *Vibrio parahaemolyticus* in locally farmed shellfish in Singapore. In this study, ampicillin, penicillin G, tetracycline resistant, and non-AMR hemolytic *V. parahaemolyticus* were surveyed and quantified in green mussel samples from different premises in the food chain (farm and retail). The occurrence data showed that 31/45 (68.9%) of farmed green mussel samples, 6/6 (100%) farm water samples, and 41/45 (91.1%) retail shellfish samples detected the presence of hemolytic *V. parahaemolyticus. V. parahaemolyticus* counts ranged from 1.6–5.9 Log CFU/g in the retail shellfish samples and 1.0–2.9 Log CFU/g in the farm water samples. AMR risk assessments (ARRA), specifically for ampicillin, penicillin G, tetracycline, and hemolytic (non-AMR) scenarios were conducted for the full farm-to-home and partial retail-to-home chains. The hemolytic ARRA scenario estimated an average probability of illness of 5.7 × 10^−3^ and 1.2 × 10^−2^ per serving for the full and partial chains, respectively, translating to 165 and 355 annual cases per total population or 2.9 and 6.2 cases per 100,000 population, respectively. The average probability of illness per year ratios for the three ARRAs to the hemolytic ARRA were 0.82, 0.81, and 0.47 (ampicillin, penicillin G, and tetracycline, respectively) for the full chain and 0.54, 0.39, and 0.09 (ampicillin, penicillin G, and tetracycline, respectively) for the partial chain. The sensitivity analysis showed that the overall cooking effect, initial concentrations of pathogenic *V. parahaemolyticus*, and harvest duration and harvest temperature were key variables influencing the risk estimates in all of the modelled ARRAs. The study findings can be used by relevant stakeholders to make informed decisions for risk management that improve food safety.

## 1. Introduction

Commonly found in marine or estuarine environments, *Vibrio parahaemolyticus* frequently colonizes and proliferates rapidly in aquatic food animals during warm seasons. First discovered in 1953, it is a foodborne pathogen that is associated with gastroenteritis caused by the consumption of raw or undercooked seafood [1]. Gastroenteritis symptoms include vomiting, diarrhea, fever, nausea, abdominal cramps, wound infection, and septicaemia [2]. Infection cases have been rising worldwide over the last few decades, while in Asia, around half of all foodborne outbreaks, particularly in Thailand, Taiwan, Japan, and several other Southeast Asian countries, were caused by this pathogen [3,4].

Not all *V. parahaemolyticus* strains can cause disease in humans. Clinical *V. parahaemolyticus* strains harboring thermostable direct hemolysin (*tdh*) genes streaked on watgasuma agar supplemented with human blood exhibit the kanagawa phenomenon where beta-emolysis is observed, while other clinical strains harboring TDH-related hemolysin (*trh*) genes exhibit alpha-hemolysis when streaked on normal human blood agar [5,6,7]. Both hemolytic reactions were also observed when clinical strains were streaked on sheep blood agar [8]. Despite the complex nature of pathogenicity in *V. parahaemolyticus*, these haemolytic reactions are indicative of virulence.

Antimicrobials are used to treat severe *Vibrio parahaemolyticus* infections, as most strains are sensitive to most antimicrobials that are critically important [9]. Antimicrobials used in aquaculture settings can not only treat such bacterial infections, but also serve as a prophylactic measure and promote growth in reared species. These beneficial properties have often led to the abusive use of antimicrobials in aquaculture settings, spurring the rapid development and dissemination of AMR in *Vibrio* spp. over the past years [10,11]. Commonly, a high proportion of environmental and clinical *V. parahaemolyticus* strains have shown resistance to the penicillin class of drugs in many countries [12,13,14,15,16]. Resistance to the tetracycline class of drugs has also been appearing in shrimp aquaculture from Hangzhou, China, and in the coastal waters in the Adriatic Sea in recent years [17,18]. Within Singapore, *Vibrio parahaemolyticus* strains with resistance to tetracycline and oxytetracycline have been consistently isolated from locally reared marine food fish [19]. The presence and dissemination of hemolytic AMR *V. parahaemolyticus* within the food chain is thus of concern for public health and food safety, as it can lead to an increased infection rate, infection severity, and treatment failure [20].

Singapore’s aquaculture industry produces around 10% of seafood consumed locally, with most of the production stemming from coastal farms utilising floating net cages along the Johor straits of Singapore [21,22]. Widely available and affordable throughout the year, green mussels are among some of the popular seafood species reared locally [22]. Studies involving the detection and quantitative enumeration of *V. parahaemolyticus* in shellfish are largely focused on oysters, shrimp, and bloody clams [23,24,25,26,27,28,29]. These studies described hemolytic *Vibrio parahaemolyticus* contamination of variable concentrations within the edible portions of the shellfish, as well as the subsequent QMRA conducted to estimate the risks associated with their consumption. However, there are still knowledge gaps regarding the detection and enumeration of AMR *V. parahaemolyticus* in shellfish in Singapore. As the consumption of shellfish contaminated by hemolytic AMR *V. parahaemolyticus* is a serious food safety risk, measures involving public health and food safety must be taken to either mitigate, diminish, or remove such risks. Antimicrobial resistance risk assessment (ARRA) tools are thus useful for pinpointing and quantifying such risks within the food chain [30,31] so as to inform risk management measures. The information derived from conducting such assessments can be used to quantify the burden of *V. parahaemolyticus* on human health through the consumption of contaminated shellfish. Subsequently, such data will be useful for crafting targeted, effective, evidence-based public health and food safety measures.

## 2. Methodology

### 2.1. Survey Data Collection for Exposure Assessment

In order to obtain the data necessary to conduct the exposure assessment section within ARRA, a survey involving a coastal open line green mussels farm and a hypermarket was undertaken. As there are a paucity of data regarding the concentration and occurrence of *Vibrio parahaemolyticus* isolated from locally farmed green mussels, this survey intended to obtain positive rates and concentrations of *V. parahaemolyticus* in green mussels for comparisons with other studies conducted for shellfish and complementing other data obtained from the literature within the model.

### 2.2. Sample Collection

#### 2.2.1. Farm and Retail Sampling

Sampling was carried out between December 2019 and March 2020. Fifteen freshly harvested green mussel samples per sampling week were procured from a green mussel farm located in the western Johor Straits. It was assumed that open line shellfish aquaculture systems in Singapore are highly similar across farms due to the relatively small aquaculture industry compared with other countries. A total of 45 freshly harvested green mussel samples were obtained over three consecutive weeks of sampling. A total of six water samples were collected from the farm, with two water samples (1 L each) collected per sampling week for three consecutive sampling weeks. Interviews to farmers were conducted to obtain data regarding antimicrobial usage during the growing stage.

Fifteen freshly harvested green mussel samples per sampling week were procured from a major retail hypermarket chiller. A total of 45 chilled green mussel samples were obtained over three consecutive weeks of sampling. Each sample was then stored in a sterile sampling bag, placed on ice, sent to the laboratory, and processed within the same day.

#### 2.2.2. Sample Processing

Each green mussel sample was weighed before processing. The shellfish were then aseptically shucked, weighed, and tested. A total of 90 shellfish meat samples from 45 freshly harvested and 45 chilled green mussel samples were analyzed for laboratory testing.

#### 2.2.3. Presumptive Vibrio Species Direct Plate Counting and Phenotypic Screening of Haemolytic Strains

Nine parts sterile 3% saline were aliquoted to one part weighed sample for an initial ten-fold dilution in a sterile stomacher bag for each sample. A stomacher Lab-blender 400 (Seward Medical, West Sussex, Worthing, UK) was used to homogenize the samples for 90 s. A filtration system using a 0.45 µM nitrocellulose membrane filter (Sigma, Steinheim am Albuch, Germany) was used to filter each 1 L water sample. Bacteria transfer from the membrane to saline was carried out by adding the membrane filter to a tube containing 10 mL 3% saline and was then vortexed for 5 min. Then, 0.1 mL of suspension mixture was aliquoted to 0.9 mL of 3% saline for all meat homogenate and filtrate samples and serial dilution was carried out for up to 10^6^ dilution. For each dilution factor, 0.1 mL of suspension mixture was spread plated on four different types of TCBS agar, supplemented with either 32 µg/mL penicillin G, 32 µg/mL ampicillin, or 16 µg/mL tetracycline, or unsupplemented TCBS agar. Duplicates were carried out for each sample at each dilution level for all of the treatment types. All TCBS plates were incubated at 37 °C for 24 h and observed for the growth of green colonies. Using sterile velveteen sheets and a replica-plating tool (VWR, Radnor, PA, USA), the plates were then replica plated onto tryptone soya agar supplemented with 5% sheep blood (Thermofisher Scientific, Waltham, MA, USA). All of the blood agar plates were incubated at 37 °C for 24 h and observed for phenotypic identification of hemolysis. Colonies exhibiting alpha or beta hemolysis were determined visually and counted for the total hemolytic population.

#### 2.2.4. Purification of Bacterial Strains and Glycerol Stocking

Several green colonies from the TCBS plates were cross-referenced to the blood agar plates and picked. Each single picked colony was then streaked onto Luria–Bertani Miller (LB) (Difco, Franklin Lakes, NJ, USA) agar supplemented with 3% NaCl and incubated at 37 °C for 24 h.

A single pure colony was picked and cultured in 3% NaCl LB broth and incubated at 37 °C for 24 h after checking for purity by observing the colony morphology. Glycerol stock was made for each bacteria isolate and stored at −80 °C for downstream 16s rRNA sequencing for bacterial identification and antimicrobial susceptibility testing.

#### 2.2.5. 16s rRNA Gene Amplification and Sequencing

Each bacteria isolate was thawed and a loopful of culture was streaked onto 3% NaCl LB agar and incubated at 37 °C for 24 h. Amplification of the full length 16s rRNA gene was carried out using the universal primers, forward primer 27F (5′-AGAGTTTGATCCTGGCTCAG-3′) and reverse primer 1492R (5-TACGGTTACCTTGTTACGACTT-3) [32]. Two to three colonies were gently touched and directly transferred to a PCR reaction mixture containing 12.5 µL 2X REDiant PCR mastermix (Axil Scientific, Singapore), 1 µL 10 µM forward primer, 1 µL 10 µM reverse primer, and 10.5 µL deionized water for colony PCR. A T100 thermocycler (Biorad, Hercules, CA, USA) was used for the PCR reaction and the cycling conditions were as follows: initial denaturation at 95 °C for 3 min, 35 cycles of denaturation at 98 °C for 10 s, annealing at 51 °C for 15 s, and extension at 72 °C for 2 min. A final extension step was performed at 72 °C for 10 min. The PCR products were analyzed using 1% agarose gel electrophoresis supplemented with GelRed (Sigma-Aldrich, Saint Louis, MO, USA). The amplified 16s rRNA gene estimated size was 1400 bp and compared with a Generuler 1 kb DNA ladder (Thermofischer, Waltham, MA, USA). Purification of the PCR product was carried out using the DNA Clean and Concentrator kit (Zymo, Irvine, CA, USA) and quantified using NanoDrop ND-100 (Thermo Scientific, Waltham, MA, USA). Purified PCR products were sent to first base, Axil Scientific for Sanger sequencing, which utilized the ABI-PRISM 31000 Genetic Analyzer system and BigDye Terminator v3.1 Cycle Sequencing kit chemistry. BioEdit Sequence Alignment Editor was used to align and combine the forward and reverse sequences results. The online *BlastN* software (http://www.ncbi.nlm.nih.gov/BLAST/, accessed 5 December 2020) version (2.11.0) was used for taxonomic identification of the sequences from the 16s rRNA gene of the bacterial isolates. Each bacterial isolate was identified according to the top hit of results with a similarity percentage ≥ 99.0%.

#### 2.2.6. Antimicrobial Susceptibility Testing

Antimicrobial susceptibility testing was carried out for all 456 bacterial isolates using the disc diffusion method. Each bacterial isolate was cultured in 5 mL of 3% NaCl supplemented LB broth at 37 °C for 24 h. The culture concentration was adjusted to 0.5 McFarland standard and a sterile swab was dipped in culture and swabbed on the entire surface of Muller–Hinton agar (Oxoid, UK) and air dried. Eight antimicrobial susceptibility test discs containing ampicillin (10 µg), ampicillin/sulbactam (10 µg/10 µg), cefotaxime (30 µg), chloramphenicol (30 µg), ciprofloxacin (5 µg), penicillin G (10 unit), sulfamethoxazole/trimethoprim (1.25 µg/23.75 µg), and tetracycline (30 µg) were placed on to the inoculated agar plate and incubated at 37 °C for 24 h. Based on the Clinical and Laboratory Standards Institute (CLSI) M45-P guideline for *Vibrio* spp., the results were interpreted as sensitive, intermediate, or resistant [33]. The CLSI M100 guideline was used as the interpretative criteria for penicillin G as this antimicrobial data were unavailable in M45-P for *Vibrio* spp. [34]. Data regarding the results of disc diffusion tests can be found in Appendix A.

### 2.3. Risk Modelling Framework

ARRAs were carried out for hemolytic *Vibrio parahaemolyticus* carrying resistance to penicillin G, ampicillin, and tetracycline using the *Codex Alimentarius* guidelines for the risk analysis of foodborne AMR pathogens [35]. ARRAs were conducted with full farm-to-home chain and partial retail-to-home chain, and the results were compared. By analyzing the ARRA variables that influence risk, intervention measures were recommended to mitigate or diminish food safety risk. The risk modelling framework is shown (Figure 1). There are several assumptions made within the framework, as follows:Green mussels harvested from the farm were processed on site before being packed on ice. Products were firstly transported to the fishery port, where they were sorted and eventually sent to the hypermarket. At the hypermarket, the seafood was packaged in plastic and placed on a chiller for display.Equal survivability fitness was assumed across different *V. parahaemolyticus* strains.Cross- and co-resistance traits of *V. parahaemolyticus* strains to the other studied resistance traits were disregarded within each ARRA for specific antimicrobial resistance.Only direct exposure from contaminated green mussels through consumption was considered. Indirect transmission modes of infection and transmission to workers in the food chain were disregarded.Variable human-host immune responses to *V. parahaemolyticus* infection were disregarded.

Data regarding the concentration and occurrences of hemolytic AMR *V. parahaemolyticus* in green mussels at the pre-harvest and retail stages were obtained through the survey. The full weight and shucked weight of the green mussels were measured to calculate the concentration of pathogenic AMR *V. parahaemolyticus*. Harvest, transport, display-related temperatures and duration times, consumer’s cooking practices, consumption trends, and statistical input variables were either obtained through local surveys or from the literature, or local data were prioritized and, if unavailable, surrogate data from nearby countries were used.

Monte Carlo sampling was carried out for all input variables with 100,000 iterations per simulation using the @RISK version 7.6 software (Pallisade Corporation, Raleigh, NC, USA) to obtain risk estimate outputs such as probability of illness per serving of green mussel, probability of infection per person per year, and estimated number of cases per year of exposed population. A total of 20 simulations were performed and the averages were obtained. Table 1 summarizes all model input parameters.

### 2.4. Hazard Identification

The microbial hazards of interest in this study were hemolytic *V. parahaemolyticus* strains with AMR traits for either penicillin G, ampicillin, or tetracycline. All strains isolated were streaked on sheep blood agar and phenotypically tested for alpha or beta hemolysis. Hemolytic strains were presumed to be clinically virulent, while strains with no hemolytic activity were presumed to be non-virulent and excluded in the ARRA.

### 2.5. Exposure Assessment

The full farm-to-home chain and the partial retail-to-home chain were modelled within the exposure assessment phase and comparative analysis between these two different chains was carried out.

#### 2.5.1. *V. parahaemolyticus* Growth Rate Modelling and Adjustment Factors

There was growth of *V. parahaemolyticus* within the green mussels as it moved along the food chain framework. The broth model by Miles et al. was used to model growth within the framework with the water activity value fixed at 0.985. There are several assumptions considered for the growth rate:

There are no differences in the growth rate of the different strains of hemolytic *V. parahaemolyticus* considered.

As the growth environment is unchanged, there is no lag phase during the harvest stage.

The growth patterns of *V. parahaemolyticus* in green mussels are identical to those of oysters, as both are marine mollusks and share biological traits.

The influence of co-infection by other pathogens such as other *Vibrio* spp. or *Aeromonas* on the growth rates of *V. parahaemolyticus* were not considered in this study.

#### 2.5.2. Hemolytic *V. parahaemolyticus* Occurrence and Concentration Levels at Pre-Harvest and Retail

The occurrence and concentration data for hemolytic AMR *V. parahaemolyticus* for green mussel samples at the pre-harvest and retail stages were obtained from the survey for the pre-harvest and retail stages and categorized under four different treatment types (hemolytic, ampicillin, penicillin G, and tetracycline).

#### 2.5.3. Parameters for Harvesting and Transportation to the Hypermarket Retailer

Singapore’s climate report and survey from the farmer was used to model the harvest duration and temperatures [38]. Harvested green mussels were packaged in nets, placed on ice, and sent to the hypermarket retailer.

#### 2.5.4. Parameters for Retail Display and Transportation to Home

Green mussels in the hypermarket retailer were wrapped in plastic packaging, placed within the chiller, and labelled as chilled seafood. Temperature variations of products within the chiller were smaller compared with the open-air display, as described by a study by Jouhara [40]. Workers in charge of the seafood counter of the hypermarket were surveyed to obtain data regarding the retail display duration. Here, 90% or the majority of green mussel purchases were made in the first 3 h, while the remaining 10% of green mussel purchases were made later in the day during the next 9 h, for up to a total of 12.5 h. Green mussels are considered perishable seafood and foul when stored in high temperatures for more than an hour, making them unsafe for consumption [46]. Hence, the transportation duration of green mussels to consumer homes was modelled as being below an hour.

#### 2.5.5. Parameters for Preparation, Cooking and Consumption of Green Mussels

Undercooking green mussels results in improper thermal inactivation of pathogenic *V. parahaemolyticus* and subsequently gastroenteritis when consumed in high doses. Based on previous studies, a minimally cooked shellfish meal has a microbial load log reduction of 0 to −2 [15,42] while a moderately cooked meal has a microbial load log reduction of −2 to −5 [42]. A highly cooked shellfish meal has a microbial load log reduction of −5 to −7 [16]. A study by US FDA also attributed that 1 in every 20 bloody clam meals were undercooked [24]. Therefore, a discrete distribution was modelled for overall cooking, with 95%, 2.5%, and 2.5% of meals highly, moderately, and minimally cooked, respectively.

### 2.6. Hazard Characterisation

#### Dose–Response Relationship

The translation of exposure to doses of pathogenic *V. parahaemolyticus* through consumption to a risk estimate of probability of illness per green mussel serving was achieved using the Beta-Poisson dose–response model from US FDA [25]. As there are knowledge gaps regarding the specific model parameters pertaining to the Singapore population, the US FDA model parameters were used. In order to account for uncertainty involving α and β model parameters, a non-parametric bootstrapping procedure was used to obtain the probability weighted selection of the paired model parameters with their maximum likelihood estimates (MLEs) [25]. The MLEs of the paired model parameters and their probabilities are described (Table 2).

Changes in occurrences at the retail stage as the green mussel moves through the farm-to-retail chain was modelled.

### 2.7. Risk Characterisation

Risk estimates regarding the estimated number of cases per year was obtained by multiplying the probability of illness per person per year with the exposed population. Key variables that highly influence the risk estimates were identified through sensitivity analysis performed in @RISK software version 7.6 (Pallisade Corporation). Using the hemolytic treatment type with the overall cooking variable as the baseline, 100,000 iterations of a random run were carried out to obtain Spearman rank correlation coefficients for the key variables.

## 3. Results

### 3.1. Haemolytic, Ampicillin, Penicillin G, and Tetracycline Resistant Vibrio parahaemolyticus Occurrence and Concentration Levels

Occurrence data for hemolytic *V. parahaemolyticus* isolated from green mussels are depicted (Table 3). The highest occurrence in the farm premise was found in hemolytic treatment, followed by tetracycline, penicillin G, and ampicillin treatment. The highest occurrence in the retail premise was found in hemolytic treatment, followed by tetracycline, penicillin G, and ampicillin treatment. While the occurrence trends were similar across the farm and retail premises, overall, there were higher occurrences in the retail premise compared with the farm premise.

Concentration data for hemolytic *V. parahaemolyticus* isolated from green mussels are also depicted (Table 3). The highest concentration levels were found for hemolytic treatment, followed by ampicillin, penicillin G, and tetracycline treatment for both the farm and retail premises. While the concentration trends were similar across the farm and retail premises, overall, there were higher concentrations in the retail premise compared with the farm premise.

### 3.2. Risk Estimate Outputs across ARRAs

The modelled occurrence and concentration data for AMR hemolytic *V. parahaemolyticus* in both the farm-to-home chain and retail-to-home chains are depicted (Table 4). All risk estimate outputs, inclusive of the average probability of illness per serving (P_ill,serving_), average probability of illness per person per year (P_ill,yearly_), and number of cases per year (N_cases_) caused by the consumption of green mussels contaminated by AMR hemolytic *Vibrio parahaemolyticus* infections across different ARRAs are depicted (Table 5). The hemolytic ARRA estimated an average P_ill,serving_ of 21,701, 921, 1.75, and 569 per 100,000 servings for minimally cooked, moderately cooked, highly cooked, and average cooked scenarios, respectively, for the farm-to-home chain. The hemolytic ARRA estimated an average P_ill,serving_ of 44,871, 4390, 19.8, and 1258 per 100,000 servings for minimally cooked, moderately cooked, highly cooked, and average cooked scenarios, respectively, for the retail-to-home chain. By comparing the highly cooked and minimally cooked scenario in the hemolytic ARRA, a reduction of >99% of P_ill,serving_ were observed for both the farm-to-home chain and retail-to-home chain.

Other ARRAs were compared to the baseline hemolytic ARRA, with other factors such as cooking extent and chain fixed as constants. The ratios of other ARRAs to the hemolytic ARRA for the average P_ill,yearly_ risk estimate are depicted (Figure 2). Overall, lower ratio values are observed for the retail-to-home chain compared with the farm-to-home.

### 3.3. Sensitivity Analysis

Key variables with the biggest influence on the variability on the P_ill,serving_ risk estimate are described for both chains (Figure 3). The top two variables, overall cooking effect and pre-harvest/retail start hemolytic *V. parahaemolyticus* concentrations in green mussels, had the biggest influence on P_ill,serving_ variability for both chains. Harvest temperature and duration variables in the farm-to-home chain and the serving size variable in both chains had a lesser influence on the variability of the risk estimates. The remaining model inputs such as retail display temperature and duration, home transport duration, retail transport temperature and duration, and farm or retail occurrences were mostly non-influential in the risk estimate results.

## 4. Discussion

### 4.1. Occurrence and Concentration Trends of Vibrio parahaemolyticus in Green Mussels

It is known that shellfish are filter feeders and are able to accumulate microorganisms from the surrounding environments into their bodies at higher concentrations. At the farm point, lower hemolytic *Vibrio parahaemolyticus* occurrence counts (68.8%, 31/45) were observed within the meat. On the other hand, higher hemolytic *V. parahaemolyticus* occurrence counts (91.1%, 41/45) were observed when the green mussels were transported to the hypermarket from the farm. These results were comparable to a study performed by Bej, in which 14 out of 19 (73.6%) oyster samples from an oyster plant contained pathogenic *V. parahaemolyticus* strains [47]. Hemolytic *V. parahaemolyticus* concentrations were also higher in green mussels obtained from the hypermarket (5.9 Log_10_CFU) compared with freshly harvested green mussels from the coastal farm (3.0 Log_10_CFU), by an approximate ~100 fold (Table 3). The drastic increase in trend could be attributed to the state of the shellfish during transportation. When alive, shellfish produce a mucosal layer that serves as a first line of barrier against infection [48]. Humoral immune responses are also mediated by multiple immune factors and hemocytes in the hemolymph that specifically detect and kill pathogens, thereby inhibiting pathogen growth within their organs [49]. During harvest, the shellfish are brought out of water and processed, which can induce stress and even death in the organism. As such, immune responses may be inhibited or even lost, allowing for uninhibited pathogen growth. In this study, shellfish samples were quickly harvested from the farm and brought to the laboratory to be processed. Therefore, the *V. parahaemolyticus* levels were more similar to those of live shellfish at the farm. In contrast, a greater time would have elapsed between the harvesting and transportation of the shellfish to the hypermarket, which resulted in the increased trend. The sharp increases in concentrations and occurrences of hemolytic *V. parahaemolyticus* in green mussels could be attributed to the harvesting duration and technique. During harvesting, green mussel shellfish need to be detached from the lines from which they are grown, debearded, and subjected to barnacle removal through a rotating drum machine. Such processes are tedious and time consuming, leading to an extended harvesting time, which averages around 7 to 8 h, where the shellfish are left at ambient air temperatures before they can be packaged and transported.

### 4.2. Comparative Analysis of ARRA Risk Estimate Outputs to Other Studies

The hemolytic ARRA, which takes into account all hemolytic *V. parahaemolyticus* and disregards their AMR traits, was used for comparison with other studies. In this study, the P_ill_serving_ risk estimate for the average cooked effect for the farm-to-home chain and partial retail-to-home chain was 5.7 × 10^−3^ and 1.3 × 10^−2^, respectively (Table 5). A study conducted by Sobrinho reported P_ill_serving_ ranging from 3.1 × 10^−4^–3.6 × 10^−3^ for raw oysters based on the season and location they were harvested in Brazil [28]. Another study by Malcolm reported risk estimates with P_ill_serving_ ranging from 5.9 × 10^−4^–8.0 × 10^−4^ in cooked bloody clams based on the retail location they were purchased in Malaysia [42]. P_ill_yearly_ in a study by Yamamoto et al. was estimated at 5.60 × 10^−4^ for the consumption of cooked bloody clams in Southern Thailand, while P_ill_serving_ in another study by Sani and her team was 4.8 × 10^−6^ for cooked tiger shrimps in Malaysia [27,29]. P_ill_serving_ ranges were also highly variable from 1.1 × 10^−5^–6.6 × 10^−4^ for the consumption of raw oysters in the United States, depending on the location and season harvested [25]. Another study involving ARRA in the consumption of grey mullet finfishes in Singapore observed P_ill_serving_ ranges from 4.52 × 10^−5^–2.85 × 10^−4^ for the hemolytic scenario, depending on the chain modelled [37]. The risk estimates from other QMRA studies on shellfish, as well as the risk estimates from the previous ARRA study on finfishes, were shown to be lower or similar in magnitude compared to the risk estimates obtained in this study. The differences in the risk estimates can be explained by differences in the biological traits between shellfish and finfishes. As the studied green mussels are primarily filter feeders that feed on suspended organic matter in the water column, such feeding patterns can result in the bio-accumulation of pathogenic *V. parahaemolyticus* to higher concentrations compared with fin fishes with different feeding patterns. The differences may also be explained by how the green mussels were handled during the harvesting period. Long periods of time of up to 8 h were modelled for the harvesting duration at ambient temperatures in the risk framework, which allowed for the growth of *Vibrio parahaemolyticus* pathogens within the shellfish up to a 126 fold. The long periods were attributed to the tedious and time-consuming processing of the green mussels during harvesting, which included debearding and removal of barnacles of the shellfish surface, which were largely not needed for other shellfish such as oysters, shrimp, and bloody clams. As the green mussels produced were not meant to be consumed raw, but instead cooked, and proper cooking of the shellfish will result in great reductions in risk through thermal inactivation, rules regarding the harvesting processes were not as strict. Another possible explanation could be due to the higher initial concentrations of *V. parahaemolyticus* within the shellfish, leading to higher risk estimates in this study. An important factor that can affect the growth of pathogenic *V. parahaemolyticus* is environmental temperature. Studies showed that *V. parahaemolyticus* reportedly grew optimally at 30–37 °C, with a maximum temperature at 44 °C and a proportional increase in growth rate with temperature from 15 to 37 °C [50,51,52]. The higher sea temperatures of around 26–32 °C might have explained the higher concentrations and occurrences of *V. parahaemolyticus* within green mussels, owing to the tropical climate in Singapore [53].

Epidemiological data regarding sporadic gastroenteritis cases caused by this pathogen are absent in Singapore, as such cases are not legally notifiable. Therefore, clinical data required to validate ARRA models were absent. In a study conducted by Gurpreet, it was found that annually, roughly 5% of the Malaysian population suffered from acute diarrhea cases of a sporadic nature, within which 3% were attributed to *V. parahaemolyticus* in another study by Bilung [54,55]. Using the surrogate data described in Malaysia, there would be 285,179 (5000 per 100,000 population) gastroenteritis cases annually in Singapore, of which 8556 cases were attributed by this pathogen through the consumption of all types of contaminated seafood. Using this surrogate data, it is thus predicted from the literature that there will be 8 gastroenteritis cases reported annually per population or 1.4 × 10^−1^ cases per 100,000, as estimated from the consumption of undercooked green mussels that are locally farmed. An average of 165 cases per year were estimated in this study, with the 5th and 95th percentile being 0 to 442 cases for the farm-to-home chain, and an average of 356 cases per year estimated, with the 5th and 95th percentile being 0 to 2809 cases for the retail-to-home chain from the consumption of green mussels (Table 5). As the 5th and 95th percentile range of cases estimated from the ARRA overlapped with the predicted number of cases, the current ARRA models were validated and found to be robust enough to assess and handle scenario changes in the model inputs.

### 4.3. Comparison Analysis among ARRAs

Risk estimate comparisons between the farm-to-home chain and retail-to-home chain were made, fixing hemolytic ARRA as the constant. P_ill_yearly_ risk estimates were 2.2-fold higher for average cooked, similar for minimally cooked, 1.6-fold higher for moderately cooked, and 6.4-fold higher for highly cooked scenario in the retail-to-home chain compared to the farm-to-home chain (Table 5). The higher risk output estimates from the retail-to-home chain were in line with the higher concentration and occurrence results obtained at the retail premise compared with the farm premise due to increased temperature and duration exposure as the green mussels progressed within the chain, consequently leading to higher risk estimates for the retail-to-home chain compared with the farm-to-home chain. The higher concentration and occurrence results at the retail premise indicate that temperature and duration are important factors that must be controlled in order to limit the growth of this pathogen through the food chain.

Risk estimate comparisons were made in the hemolytic ARRA between different cooking scenarios, fixing the chain as the constant. P_ill_yearly_ risk estimates were 120-fold higher in the minimally cooked, 49-fold higher for the moderately cooked, and 5.2-fold higher for the average cooked compared with the highly cooked scenario within the farm-to-home chain (Table 5). P_ill_yearly_ risk estimates were 19-fold higher in the minimally cooked, 12-fold higher for the moderately cooked, and 1.7-fold higher for the average cooked compared with the highly cooked scenario within the retail-to-home chain (Table 5). The reduction in risk associated with proper cooking showed that this is a key variable influencing risk estimates and food safety.

P_ill_yearly_ risk estimates ratio comparisons were made for ampicillin, penicillin G, and tetracycline ARRAs to hemolytic ARRA, fixing the cooking extent as the average cooked. For the farm-to-home chain, P_ill_yearly_ ratios for the ampicillin and penicillin G ARRAs were higher compared with tetracycline ARRA, with ratios of 8.2 × 10^−1^, 8.2 × 10^−1^, and 4.8 × 10^−1^ for ampicillin, penicillin G, and tetracycline ARRAs, respectively (Figure 2). In freshly harvested green mussels, risk estimates from ampicillin and penicillin G ARRA were relatively lower compared with hemolytic ARRA, while risk estimates from tetracycline ARRA were relatively much lower compared with hemolytic ARRA. The trend remained unchanged for the retail-to-home chain, with P_ill_yearly_ ratios of 5.4 × 10^−1^, 3.9 × 10^−1^, and 9.5 × 10^−2^ for the ampicillin, penicillin G, and tetracycline ARRAs, respectively. There was an overall drop in ratio values for the retail-to-home chain compared with the farm-to-home chain, owing to a higher risk of illness from hemolytic non-AMR *V. parahaemolyticus* in the retail-to-home chain compared with the farm-to-home chain. The observed trend within both the full and partial chain indicated that the majority of gastroenteritis cases of up to ~82% for the farm-to-home chain and ~39–54% for the retail-to-home chain were caused by hemolytic strains carrying either or both ampicillin and penicillin G resistanc, while a smaller number of gastroenteritis cases of up to ~48% for the farm-to-home chain and ~9.5% for the retail-to-home chain would be caused by hemolytic tetracycline-resistant *V. parahaemolyticus* strains. P_ill_yearly_ risk estimates were the highest for ampicillin ARRA, followed by penicillin G ARRA and tetracycline ARRA in both chains. The results show that consuming green mussels contaminated by ampicillin or penicillin G-resistant *V. parahaemolyticus* resulted in a greater risk of gastroenteritis compared with tetracycline-resistant *V. parahaemolyticus*.

### 4.4. Sensitivity Analysis of Intervention Measures

Sensitivity analysis of the hemolytic ARRA for both chains showed that overall cooking effect, initial concentrations of the pathogen at pre-harvest and retail start stages, and harvest temperatures and duration were the main key variables that had the biggest influence on risk estimates (Figure 3). The cooking process, which is associated with how the consumer handles food, is negatively correlated with risk, while initial concentrations of the pathogen were tied to aquaculture farm practices and seafood handling during harvest and transport to retail and were positively correlated with risk. Evidence showed that pathogen growth occurred throughout the food chain, with the largest increase in concentrations occurring during the harvesting stage. Measures can then be taken to reduce pathogen loads within the green mussels during harvest and prior to retail display. Harvesting duration should be reduced as much as possible through the employment of more labor or the use automated machinery to speed up the harvesting and post-harvesting processes [23]. Other intervention measures to reduce initial pathogen concentrations include depuration, relaying, mild heat, hot and cold-water shock, and high hydrostatic-pressure treatments. These treatments have been shown to be effective at reducing the pathogen load in oysters [23]. Pertaining to consumer’s handling of food, estimated risks are the highest when the food is inadequately cooked. Thus, ensuring that the shellfish is thoroughly cooked prior to consumption is critical to significantly reduce risks. Other intervention measures that target retail display temperatures and duration, as well as home or retail transportation temperatures and durations, may help in reducing overall risk, but have only a marginal effect in reducing risk [23].

A limitation in the study is that cross contamination was not modelled. Cross contamination can occur within the entire farm-to-home chain, such as processing and packaging of the green mussels during harvesting, packaging of the seafood prior to retail display, and improper food handling by the consumers. In a study by de Jong, up to 40 to 60% foodborne illnesses were reportedly associated with improper food handling and surface cross contamination [56]. With such great impacts, cross-contamination can potentially play in influencing risk estimates, more data are required to understand the extent and probability extent of such events within the modelling framework, such as the harvesting, handling, and transportation of the green mussels to the hypermarket and the handling of seafood products of other sources on a common counter prior to retail display. One other limitation is that co-infection with other bacterial pathogens was not modelled. Hibbing and his team highlighted that most environments host a range of microbial species, in which bacteria of different species can compete or cooperate with each other for space and resources through direct or indirect means [57]. Such interactions, especially in co-infection scenarios, can influence the growth rate of *V. parahaemolyticus* in green mussels, which influences risk estimates downstream. As such, more data regarding the bacterial microbial community and their population growth dynamics in green mussels are needed to understand the impacts co-infection have on risk estimates. Another limitation is the relatively small sample size due to the relatively smaller population of farm production and the available samples. As the generation of exposure assessment data inputs requires conducting such experiments to consider meaningful ARRAs, such small sample sizes can contribute to uncertainty in the model. In spite of these limitations, the risk estimate data obtained can be used by relevant stakeholders in the local aquaculture industry to make informed decisions concerning food safety.

As part of future works, a deeper analysis of the antimicrobial resistance in *V. parahaemolyticus* isolates obtained from this study can be carried out through whole genome sequencing to understand the genotypic traits that contribute to their virulence and AMR determinants [58]. Pathogenic strains can be grouped or segregated refined through profiling of all virulence and AMR genetic determinants, thus allowing strain specific adjustments to the exposure assessment and dose–response models based on their growth/survival ability and their virulence traits. Data inputs regarding the magnitude of exposure at consumption and the estimation of probability of illness can thus be improved, thereby refining the model’s prediction accuracy [59].

## 5. Conclusions

This study is the first to report the concentrations and occurrences of hemolytic AMR *V. parahaemolyticus* from locally farmed green mussels in Singapore. The study showed that this pathogen was detected from green mussel samples derived from either the farm or hypermarket. AMR *Vibrio parahaemolyticus* concentrations and occurrences were higher in green mussels obtained from the hypermarket compared with green mussels obtained from the farm. This study also quantified the risks caused by this pathogen found in green mussels. Risk estimate outputs were lower in the farm-to-home chain compared with the retail-to-home chain. Furthermore, AMP and PENG-resistant *V. parahaemolyticus* strains posed a higher risk for gastroenteritis compared with TET-resistant strains. Key model variables that greatly influenced risk were identified and highlighted through sensitivity analysis, such as the initial concentrations of the pathogen within the green mussels at the retail or pre-harvest stages, cooking effect, and the harvest temperature and duration, allowing risk management measures to be made to diminish or mitigate risk. Through this study, relevant stakeholders such as aquaculture farmers, governmental bodies, and consumers of seafood can utilize the risk data generated to make better informed decision-making processes based on scientific evidence. These can include improving aquaculture husbandry practices, improving policy planning, and increasing education for safer food handling for aquaculture farmers, governmental bodies, and consumers, respectively.

## Figures and Tables

**Figure 1 microorganisms-11-01498-f001:**
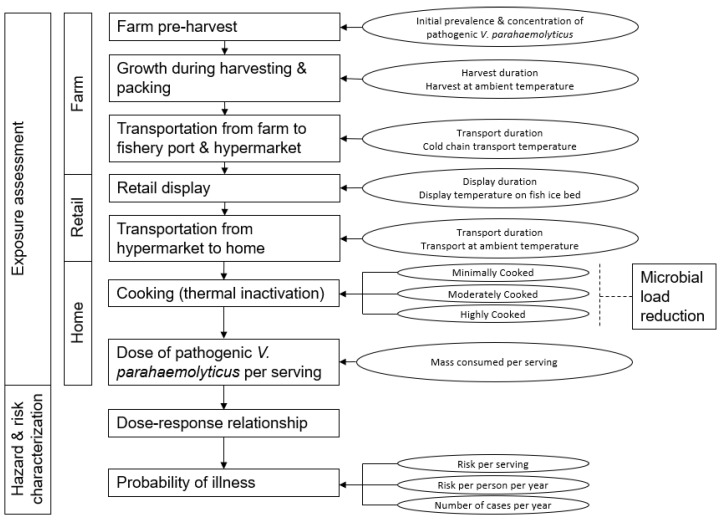
Risk modelling framework depicting ARRA of hemolytic *V. parahaemolyticus* isolated from green mussels.

**Figure 2 microorganisms-11-01498-f002:**
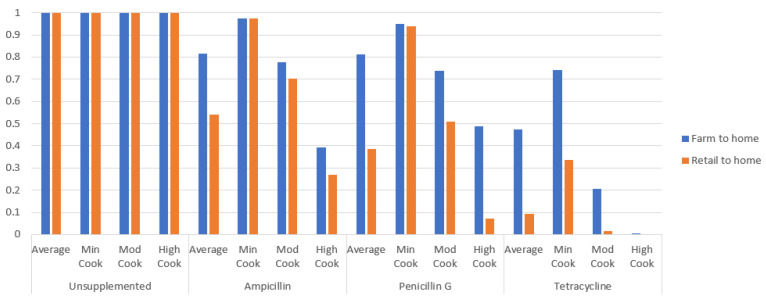
Comparative ratios of average P_ill,yearly_ of ARRAs to the hemolytic ARRA.

**Figure 3 microorganisms-11-01498-f003:**
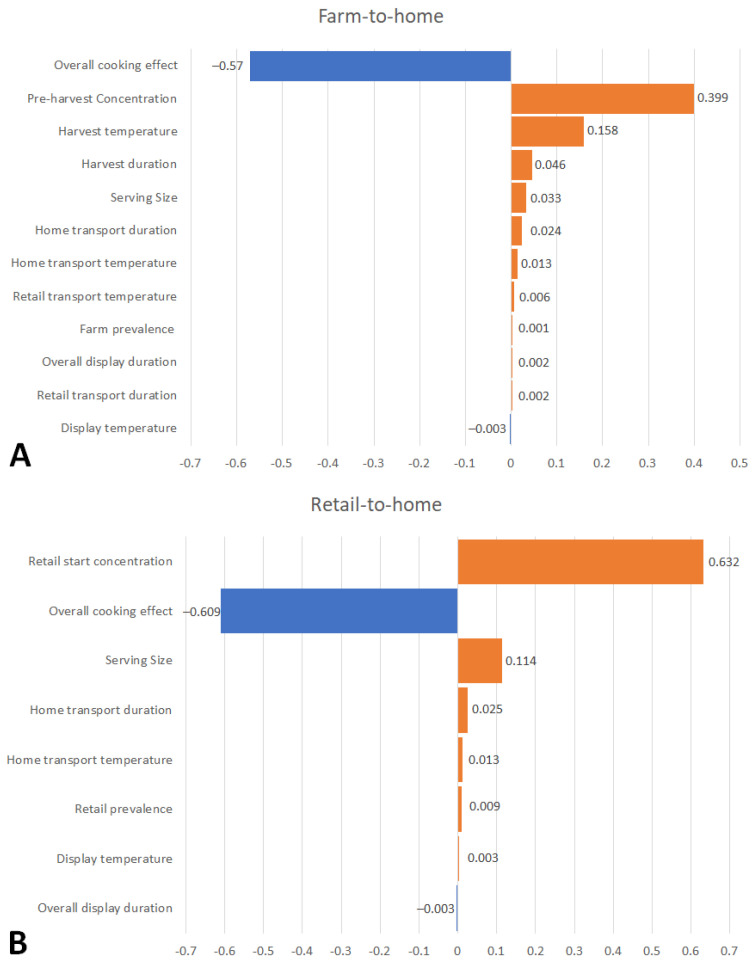
Tornado charts depicting Spearman rank correlation coefficients of input variables influencing P_ill,serving_ risk estimates in hemolytic ARRA: (**A**) Farm-to-home chain and (**B**) retail-to-home chain.

**Table 1 microorganisms-11-01498-t001:** All ARRA model input parameters for *V. parahaemolyticus* (Vp) in green mussels.

Symbol	Description	Equation	References
**Exposure assessment**			
**Growth rate equations**			
$\sqrt{K}$	Growth rate in broth model (Log_10_/min)	$\frac{0.035634\left(T-278.5\right)\left[1-exp(0.3403\left(T-319.6\right)\right)]*\sqrt{\left(a_{w}-0.921\right)\left[1-exp(263.64\left(a_{w}-0.998\right)\right)]}}{\sqrt{ln(10)}}$	[36]
$K_{}$	Growth rate (Log_10_/h)	${\left(\sqrt{K}\right)}^{2}\times 60$	-
$K_{\mathrm{AD}}$	Growth rate adjustment	$\frac{K}{\mathrm{RiskTriang}\left(3,4,5\right)}$	[25]
**Initial occurrence and concentration equations**			
$P_{{\mathrm{patho}}_{\mathrm{farm}}}$	Occurrence of haemolytic *Vp*	RiskBeta (positives + 1, negatives + 1)	[37]
$Vp_{\mathrm{pre}-\mathrm{harvest}}\;$ ^#^	Total concentration of *Vp* in shellfish (Log_10_/g)	Log[CFU / g meat]	-
Growth during harvest			
$t_{\mathrm{harvest}}$	Harvest time (h)	RiskTriang (7, 7.5, 8)	Author’s input
$T_{\mathrm{harvest}}$	Harvest temperature (K)	RiskPert (299.05, 301.55, 305.45)	[38]
$Vp_{\mathrm{post}-\mathrm{harvest}}$	Concentration of *Vp* (Log_10_/g)	$Vp_{\mathrm{pre}-\mathrm{harvest}}+\left[Vp_{\mathrm{pre}-\mathrm{harvest}}\times \left(K_{\mathrm{AD}}\times t_{\mathrm{harvest}}\right)\right]$	-
**Growth during transport to retail**			
$t_{F\to R}$	Transport time (h)	RiskUniform (13.5, 14.5)	Author’s input
$T_{F\to R}$	Transport temperature (K)	RiskPert (276.15, 279.15, 282.15)	[39]
$Vp_{\mathrm{retail}\;\mathrm{start}}$	Concentration of *Vp*	$Vp_{\mathrm{post}-\mathrm{harvest}}+\left[Vp_{\mathrm{post}-\mathrm{harvest}}\times \left(K_{\mathrm{AD}}\times t_{F\to R}\right)\right]$	-
**Growth during retail display**			
$t_{\mathrm{retail}-90\%}$	Display time for majority of purchases (h)	RiskUniform (0.5, 3.5)	Author’s input
$t_{\mathrm{retail}-10\%}$	Display time for remaining purchases (h)	RiskPert (3.5, 5.5, 12.5)	Author’s input
$t_{\mathrm{overall}\;\mathrm{retail}}$	Overall display time (h)	$\mathrm{RiskDiscrete}(t_{\mathrm{retail}-90\%}:t_{\mathrm{retail}-10\%})$	Author’s input
$T_{\mathrm{retail}}$	Display temperature (K)	RiskPert (275.85, 278.15, 279.15)	[40]
$Vp_{\mathrm{retail}\;\mathrm{end}}$	Concentration of *Vp* (Log_10_/g)	$Vp_{\mathrm{retail}\;\mathrm{start}}+\left[Vp_{\mathrm{retail}\;\mathrm{start}}\times \left(K_{\mathrm{AD}}\times t_{\mathrm{overall}\;\mathrm{retail}}\right)\right]$	-
**Growth during transport to home**			
$t_{R\to H}$	Transport time (h)	RiskTriang (0.25, 0.5, 0.75)	[41]
$T_{R\to H}$	Transport temperature (K)	RiskPert (299.05, 301.55, 305.45)	[38]
$Vp_{\mathrm{home}}$	Concentration of *Vp* (Log_10_/g)	$Vp_{\mathrm{retail}\;\mathrm{end}}+\left[Vp_{\mathrm{retail}\;\mathrm{end}}*\left(K_{\mathrm{AD}}\times t_{R\to H}\right)\right]$	-
**Preparation and cooking**			
${\mathrm{Cooking}}_{\mathrm{minimal}}$	Minimally cooked scenario (Log_10_/g)	RiskUniform (0, −2)	[15,42]
${\mathrm{Cooking}}_{\mathrm{moderate}}$	Moderately cooked scenario (Log_10_/g)	RiskUniform (−2,−5)	[42]
${\mathrm{Cooking}}_{\mathrm{highly}}$	Highly cooked scenario (Log_10_/g)	RiskUniform (−5,−7)	[16]
${\mathrm{Cooking}}_{\mathrm{Overall}}$	Overall cooked scenario (Log_10_/g)	Cooking_minimal_ −2.5% Cooking_moderate_ −2.5% Cooking_highly_ −95% ${\mathrm{Cooking}}_{\mathrm{overall}}=\mathrm{RiskDiscrete}({\mathrm{Cooking}}_{\mathrm{minimal}}:{\mathrm{Cooking}}_{\mathrm{moderate}}:{\mathrm{Cooking}}_{\mathrm{highly}})$	[24]
$Vp_{{\mathrm{dose}}_{i}}\;$ ^&^	Concentration of *Vp* (Log_10_/g)	$Vp_{\mathrm{home}}+{\mathrm{Cook}}_{i}$	-
Serv	Serving Size (grams)	RiskTriang (0, 17.2, 90.47)	[43]
d	Dose (CFU)	$10^{Vp_{{\mathrm{dose}}_{i}}}*Serv$	-
**Hazard characterisation**			
BP	Beta Poisson dose-response	$1-{\left(1+\frac{d}{\beta }\right)}^{-\alpha }$	[25]
$P_{{\mathrm{patho}}_{F\to R}}\;$ ^@^	Occurrence change from farm to retail	$P_{{\mathrm{patho}}_{\mathrm{farm}}}+\left(1-P_{{\mathrm{patho}}_{\mathrm{farm}}}\right)\times \left(P_{{\mathrm{patho}}_{\mathrm{retail}}}-P_{{\mathrm{patho}}_{\mathrm{farml}}}\right)$	-
$P_{\mathrm{ill},\mathrm{serving}}$	Probability of illness per serving	$BP\times P_{{\mathrm{patho}}_{i}}\;$ ^@^	-
**Risk characterisation**			
$Pol_{\mathrm{Sin}\mathrm{gapore}}$	Singapore’s population	5,703,569	[44]
$Pol_{\mathrm{fish}}$	Population proportion consuming shellfish	0.937	[45]
$Pol_{\mathrm{green}\;\mathrm{mussel}}$	Population proportion consuming green mussel	$\frac{30}{33,475}=8.962\times 10^{-4}$	Author’s input
$Pol_{\mathrm{exposed}}$	Exposed population	$Pol_{\mathrm{Sin}\mathrm{gapore}}\times Pol_{\mathrm{shellfish}}\times Pol_{\mathrm{green}\;\mathrm{mussel}}=4790$	-
n	Number of meals per week	RiskNormal (8.29, 8.323)	[45]
$P_{\mathrm{ill},\mathrm{yearly}}$	Probability of illness per person per year	$1-{\left(1-P_{\mathrm{ill},\mathrm{serving}}\right)}^{\left(n\times 52\right)}$	-
$N_{\mathrm{cases}}$	Cases per year	$P_{\mathrm{ill},\mathrm{yearly}}\times Pol_{\mathrm{exposed}}$	-

^#^ Values of Vp_pre-harvest_ were based on the four different ARRA scenarios: hemolytic, ampicillin, penicillin G, and tetracycline. ^&^
i = minimal, moderate, highly, or overall cooking scenarios. ^@^
$P_{{\mathrm{patho}}_{F\to R}}$ or $P_{{\mathrm{patho}}_{\mathrm{Retail}}}$ scenarios.

**Table 2 microorganisms-11-01498-t002:** MLEs of α and β paired model parameters for Beta-Poisson dose–response model and their corresponding probability weights.

FDA Model	α	β	Probability Weight
1	1.47 × 10^6^	3.53 × 10^14^	3.40 × 10^−4^
2	1.26 × 10^7^	7.20 × 10^14^	4.12 × 10^−3^
3	6.37 × 10^2^	1.65 × 10^10^	2.06 × 10^−2^
4	3.58 × 10^1^	5.42 × 10^8^	5.49 × 10^−2^
5	2.08 × 10^1^	1.99 × 10^8^	8.23 × 10^−2^
6	1.49 × 10^1^	8.78 × 10^7^	6.58 × 10^−2^
7	1.06 × 10^1^	2.99 × 10^7^	2.20 × 10^−2^
8	3.89	2.28 × 10^8^	6.90 × 10^−4^
9	1.31	2.93 × 10^7^	8.23 × 10^−3^
10	5.20 × 10^−1^	3.61 × 10^6^	4.12 × 10^−2^
11	4.70 × 10^−1^	1.50 × 10^6^	1.10 × 10^−1^
12	6.00 × 10^−1^	1.31 × 10^6^	1.65 × 10^−1^
13	1.00	1.80 × 10^6^	1.32 × 10^−1^
14	8.59	1.30 × 10^7^	4.39 × 10^−2^
15	1.50 × 10^−1^	2.33 × 10^5^	3.40 × 10^−4^
16	1.90 × 10^−1^	2.29 × 10^5^	4.12 × 10^−3^
17	2.50 × 10^−1^	2.36 × 10^5^	2.06 × 10^−2^
18	3.20 × 10^−1^	2.57 × 10^5^	5.49 × 10^−2^
19	4.30 × 10^−1^	3.04 × 10^5^	8.23 × 10^−2^
20	6.90 × 10^−1^	4.34 × 10^5^	6.58 × 10^−2^
21	6.92	4.49 × 10^6^	2.20 × 10^−2^

**Table 3 microorganisms-11-01498-t003:** Occurrence (%) and concentration (S.E.) of hemolytic *V. parahaemolyticus* isolates in green mussel and farm water samples from a coastal marine farm and a hypermarket in Singapore. AMP-R: ampicillin resistant; PENG-R: Penicillin G resistant; TET-R: tetracycline resistant. Percentages are calculated with the denominator as the sample total or water sample total.

VP	Marine Coastal Farm		Hypermarket		Farm Water
	Sample Total	Meat	Sample Total	Meat	
Occurrence: haemolytic	45	31/45 (69)	45	41/45 (91)	6/6 (100)
Mean concentration: haemolytic	-	3.0 (0.10)	-	5.9 (0.12)	2.8 (0.17)
Occurrence: AMP-R	45	14/45 (31)	45	30/45 (67)	4/6 (67)
Mean concentration: AMP-R	-	2.8 (0.13)	-	5.3 (0.13)	2.7 (0.25)
Occurrence; PENG-R	45	15/45 (33)	45	34/45 (76)	5/6 (83)
Mean concentration: PENG-R	-	2.7 (0.17)	-	4.8 (0.11)	2.3 (0.34)
Occurrence: TET-R	45	25/45 (56)	45	40/45 (89)	5/6 (83)
Mean concentration: TET-R	-	1.6 (0.081)	-	2.5 (0.075)	1.0 (0.075)

**Table 4 microorganisms-11-01498-t004:** Occurrence and concentration changes of hemolytic *V. parahaemolyticus* along the both chains, where 20 runs and 100,000 iterations per run are performed using Monte Carlo simulations to obtain the. concentration and occurrence data. The 5th and 95th percentages are represented in brackets.

			Farm-To-Home				Retail-To-Home			
			Haemolytic	Ampicillin	Penicillin G	Tetracycline	Haemolytic	Ampicillin	Penicillin G	Tetracycline
	Occurrence (Farm/Retail)		6.8 × 10^−1^ (5.7 × 10^−1^, 7.9 × 10^−1^)	3.2 × 10^−1^ (2.1 × 10^−1^, 4.3 × 10^−1^)	3.4 × 10^−1^ (2.3 × 10^−1^, 4.6 × 10^−1^)	5.5 × 10^−1^ (4.3 × 10^−1^, 6.7 × 10^−1^)	8.9 × 10^−1^ (8.1 × 10^−1^, 9.6 × 10^−1^)	6.6 × 10^−1^ (5.4 × 10^−1^, 7.7 × 10^−1^)	7.5 × 10^−1^ (6.4 × 10^−1^, 8.4 × 10^−1^)	8.7 × 10^−1^ (7.9 × 10^−1^, 9.4 × 10^−1^)
Concentration (LogCFU/g)	Farm	Pre-harvest	3.0 (2.1, 3.9)	2.8 (2.0, 3.6)	2.7 (1.6, 3.8)	1.60 (9.3 × 10^−1^, 2.3)	* N.A.	N.A.	N.A.	N.A.
		Post-harvest	5.1 (4.1, 6.2)	4.9 (4.0, 5.9)	4.8 (3.6, 6.0)	3.7 (2.9, 4.5)	N.A.	N.A.	N.A.	N.A.
	Retail	Retail-start	5.1 (4.1, 6.2)	5.0 (4.0, 5.9)	4.8 (3.7, 6.0)	3.7 (2.9, 4.6)	5.9 (4.6, 7.1)	5.3 (4.1, 6.5)	4.8 (3.8, 5.9)	2.5 (1.7, 3.3)
		Retail-end	5.2 (4.1, 6.2)	5.0 (4.0, 5.9)	4.8 (3.7, 6.0)	3.7 (2.9, 4.6)	5.9 (4.6, 7.1)	5.3 (4.2, 6.5)	4.8 (3.8, 5.9)	2.5 (1.7, 3.3)
		Home	5.3 (4.2, 6.3)	5.1 (4.1, 6.1)	5.0 (3.8, 6.1)	3.8 (3.0, 4.7)	6.0 (4.7, 7.3)	5.5 (4.3, 6.7)	5.0 (3.9, 6.0)	2.6 (1.8, 3.4)
	Cooking	Average	2.5 × 10^−1^ (0, 1.3)	1.9 × 10^−1^ (0, 1.0)	1.9 × 10^−1^ (0, 1.1)	8.7 × 10^−2^ (0, 0)	5.6 × 10^−1^ (0, 2.2)	3.2 × 10^−1^ (0, 1.6)	1.8 × 10^−1^ (0, 9.7 × 10^−1^)	4.3 × 10^−2^ (0, 0)
		Minimally cooked	4.3 (2.9, 5.7)	4.1 (2.7, 5.4)	4.0 (2.5, 5.5)	2.8 (1.6, 4.1)	5.0 (3.4, 6.6)	4.5 (3.0, 6.0)	4.0 (2.6, 5.4)	1.6 (3.9 × 10^−1^, 2.8)
		Moderately cooked	1.8 (4.1 × 10^−2,^, 3.5)	1.6 (0, 3.3)	1.5 (0, 3.3)	6.1 × 10^−1^ (0, 2.0)	2.5 (6.2 × 10^−1^, 4.4)	2.0 (1.4 × 10^−1^, 3.8)	1.5 (0, 3.2)	9.9 × 10^−2^ (0, 7.0 × 10^−1^)
		Highly cooked	1.0 × 10^−1^ (0, 7.0 × 10^−1^)	5.5 × 10^−2^ (0, 4.4 × 10^−1^)	5.9 × 10^−2^ (0, 4.7 × 10^−1^)	2.4 × 10^−4^ (0, 0)	4.0 × 10^−1^ (0, 1.6)	1.7 × 10^−1^ (0, 1.0)	4.9 × 10^−2^ (0, 4.0 × 10^−1^)	0 (0, 0)

* For retail-to-home chain, green mussels are not sampled from the farm but only from retail, hence non-applicable.

**Table 5 microorganisms-11-01498-t005:** Comparative analysis of risk estimates across all ARRAs, where 20 runs and 100,000 iterations per run are performed using Monte Carlo simulations to obtain all of the average risk estimates. The 5th and 95th percentages are represented in brackets.

		Farm-To-Home			Retail-To-Home		
		P_ill,serving_	P_ill,yearly_	N_cases_	P_ill,serving_	P_ill,yearly_	N_cases_
Haemolytic	Average	5.7 × 10^−3^ (0, 2.9 × 10^−4^)	3.4 × 10^−2^ (0, 9.2 × 10^−2^)	1.7 × 10^2^ (0, 4.4 × 10^2^)	1.3 × 10^−2^ (0, 2.7 × 10^−3^)	7.4 × 10^−2^ (0, 5.9 × 10^−1^)	3.6 × 10^2^ (0, 2.8 × 10^3^)
	Minimally cooked	2.2 × 10^−1^ (3.5 × 10^−3^, 6.6 × 10^−1^ )	8.0 × 10^−1^ (0, 1)	3.9 × 10^3^ (0, 4.8 × 10^3^)	4.5 × 10^−1^ (1.6 × 10^−2^, 8.9 × 10^−1^)	8.3 × 10^−1^ (0, 1)	4.0 × 10^3^ (0, 4.8 × 10^3^)
	Moderately cooked	9.2 × 10^−3^ (1.6 × 10^−6^, 4.4 × 10^−2^)	3.3 × 10^−1^ (0, 1)	1.6 × 10^3^ (0, 4.8 × 10^3^)	4.4 × 10^−2^ (2.6 × 10^−5^, 2.6 × 10^−1^)	5.1 × 10^−1^ (0, 1)	2.4 × 10^3^ (0, 4.8 × 10^3^)
	Highly cooked	1.8 × 10^−5^ (0, 7.5 × 10^−5^)	6.7 × 10^−3^ (0, 3.0 × 10^−2^)	3.2 × 10^1^ (0, 1.4 × 10^2^)	2.0 × 10^−4^ (0, 6.7 × 10^−4^)	4.3 × 10^−2^ (0, 2.5 × 10^−1^)	2.1 × 10^2^ (0, 1.2 × 10^3^)
Haemolytic and AMP-R	Average	3.4 × 10^−3^ (0, 1.2 × 10^−4^)	2.8 × 10^−2^ (0, 3.9 × 10^−2^)	1.4 × 10^2^ (0, 1.9 × 10^2^)	6.1 × 10^−3^ (0, 5.0 × 10^−4^)	4.0 × 10^−2^ (0, 1.5 × 10^−1^)	1.9 × 10^2^ (0, 7.2 × 10^2^)
	Minimally cooked	1.3 × 10^−1^ (1.9 × 10^−3^, 4.5 × 10^−1^)	7.8 × 10^−1^ (0, 1)	3.8 × 10^3^ (0, 4.8 × 10^3^)	2.3 × 10^−1^ (3.9 × 10^−3^, 6.2 × 10^−1^)	8.1 × 10^−1^ (0, 1)	3.9 × 10^3^ (0, 4.8 × 10^3^)
	Moderately cooked	4.2 × 10^−3^ (0, 1.9 × 10^−2^)	2.5 × 10^−1^ (0, 1)	1.2 × 10^3^ (0, 4.8 × 10^3^)	1.3 × 10^−2^ (3.6 × 10^−6^, 6.8 × 10^−2^)	3.6 × 10^−1^ (0, 1)	1.7 × 10^3^ (0, 4.8 × 10^3^)
	Highly cooked	6.2 × 10^−6^ (0, 2.8 × 10^−5^)	2.6 × 10^−3^ (0, 9.9 × 10^−3^)	1.3 × 10^1^ (0, 4.8 × 10^1^)	3.4 × 10^−5^ (0, 1.3 × 10^−4^)	1.1 × 10^−2^ (0, 5.2 × 10^−2^)	5.5 × 10^1^ (0, 2.5 × 10^2^)
Haemolytic and PENG-R	Average	3.5 × 10^−3^ (0, 1.4 × 10^−4^)	2.8 × 10^−2^ (0, 4.4 × 10^−2^)	1.4 × 10^2^ (0, 2.1 × 10^2^)	4.1 × 10^−3^ (0, 1.4 × 10^−4^)	2.9 × 10^−2^ (0, 4.5 × 10^−2^)	1.4 × 10^2^ (0, 2.2 × 10^2^)
	Minimally cooked	1.3 × 10^−1^ (1.1 × 10^−3^, 4.9 × 10^−1^)	7.6 × 10^−1^ (0, 1)	3.7 × 10^3^ (0, 4.8 × 10^3^)	1.6 × 10^−1^ (1.7 × 10^−3^, 5.8 × 10^−1^)	7.8 × 10^−1^ (0, 1)	3.7 × 10^3^ (0, 4.8 × 10^3^)
	Moderately cooked	4.8 × 10^−3^ (0, 2.1 × 10^−2^)	2.4 × 10^−1^ (0, 1)	1.2 × 10^3^ (0, 4.8 × 10^3^)	5.1 × 10^−3^ (0, 2.3 × 10^−2^)	2.6 × 10^−1^ (0, 1)	1.2 × 10^3^ (0, 4.8 × 10^3^)
	Highly cooked	8.5 × 10^−6^ (0, 3.2 × 10^−5^)	3.3 × 10^−3^ (0, 1.1 × 10^−2^)	1.6 × 10^1^ (0, 5.3 × 10^1^)	7.6 × 10^−6^ (0, 3.2 × 10^−5^)	3.1 × 10^−3^ (0, 1.0 × 10^−2^)	1.5 × 10^1^ (0, 5.1 × 10^1^)
Haemolytic and TET-R	Average	7.2 × 10^−4^ (0, 0)	1.7 × 10^−2^ (0, 0)	7.9 × 10^1^ (0, 0)	7.2 × 10^−5^ (0, 0)	7.1 × 10^−3^ (0, 0)	3.4 × 10^1^ (0, 0)
	Minimally cooked	2.8 × 10^−2^ (1.6 × 10^−4^, 1.4 × 10^−1^)	5.9 × 10^−1^ (0, 1)	2.8 × 10^3^ (0, 4.8 × 10^3^)	2.8 × 10^−3^ (1.1 × 10^−5^, 1.3 × 10^−2^)	2.7 × 10^−1^ (0, 1)	1.3 × 10^3^ (0, 4.8 × 10^3^)
	Moderately cooked	3.1 × 10^−4^ (0, 1.3 × 10^−3^)	6.6 × 10^−2^ (0, 4.2 × 10^−1^)	3.2 × 10^2^ (0, 2.0 × 10^3^)	1.8 × 10^−5^ (0, 8.5 × 10^−5^)	7.1 (0, 3.4 × 10^−2^)	3.4 × 10^1^ (0, 1.6 × 10^2^)
	Highly cooked	3.4 × 10^−8^ (0, 0)	1.5 × 10^−5^ (0, 0)	7.3 × 10^−2^ (0, 0)	0 (0, 0)	0 (0, 0)	0 (0, 0)

## Data Availability

https://www.dropbox.com/scl/fo/npiq159jmrycfv97tr91q/h?dl=0&rlkey=t61rebtteg0bv1zalc5np674e.

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
