# Peer review of "Antimicrobial Resistance Risk Assessment of Vibrio parahaemolyticus Isolated from Farmed Green Mussels in Singapore"

_microorganisms, 2023, doi:10.3390/microorganisms11061498_

Round 1

Reviewer 1 Report

The manuscript entitled "Antimicrobial Resistance Risk Assessment of Vibrio parahaemolyticus Isolated from 

Farmed Green Mussels in Singapore" presented  a good study regarding the Risk Assessment of Vibrio 

parahaemolyticus from farm to table. The overall experiment design is good. However, there are a number of issues might preclude the acceptance of this manuscript

Major comments:

1. the title is about Antimicrobial Resistance Risk Assessment, however the majority of the content is regarding 

Risk Assessment of Vibrio parahaemolyticus, the focus is not clear.

2. haemolytic V. parahaemolyticus: which genes cause haemolytic needs to be clear. Whether Vibrio 

parahaemolyticus strains used in this study  harbor thermostable direct hemolysin (TDH) or TDH-related hemolysin 

(TRH) remains unclear.

Introduction:

line 80: there are a number of  QMRA work on Vibrio parahaemolyticus in shellfish, but author did not mentioned 

any of them

line 86: the details of Antimicrobial resistance risk assessment (ARRA) tools needs to be clear.

Methods:

line 102: the location of the farm needs to be clear.

Discussion:

author did not mentioned other Risk Assessment results of Vibrio parahaemolyticus from previous studies. Besides, the average probability of illness per year also need to compare with the results from previous studies. 

Minor editing of English language required

Reviewer 2 Report

Dear Authors,

the Article “Antimicrobial Resistance Risk Assessment of Vibrio parahaemolyticus Isolated from Farmed Green Mussels in Singapore” (Microorganisms-2376101), describes an antimicrobial risk assessment of Vibrio parahaemolyticus, one of the main causes of gastroenteritis in human associated with bivalve consumption.

Introduction should be improved regarding the local rules for antimicrobial treatments in aquaculture that could be than used to implement the discussion to comment the antimicrobial resistance data.

Line 59: Please, explain which are the local rules for antimicrobial treatments and which are the molecules allowed for aquaculture; moreover, the time of suspension to be respected before consumption of bivalve if treatments are applied.

I have concern regarding Methodology and the assumption done at paragraph 2.5.1. V. parahaemolyticus growth rate modelling and adjustment factors: you assumed that there are no differences in the growth rate of different strains of V. parahaemolitycus but the eventual presence of co-infection and how the growth rate could be influenced by the interaction with other Vibrio species or Aeromonas (already reported in literature) have not been considered.

The same in the discussion; at line 524 it is reported that a limitation in the study is that cross-contamination is not modelled. If the variable co-infection is not modelled, you should underline that this represents another limit of the study and how this could influence the results obtained.

Moreover, I have the following minor concerns:

The text needs to be revised for the presence of a lot of typo errors (for example line 105: forty five; line 147: 16s; paragraph 2.2.4 16. s rRNA; line 195 Figure 1 and not Figure 3; line 219 amend or; line 368 remove the bold style from the caption of Figure 3; line 378 insert  a full-stop between the two sentences; line 386 emoyces), and scientific name of bacteria should be reported using italics all through the text (line 95; line 211) and the correct name of the bacterium is Vibrio parahaemolyticus and not Vibrio parahemolyticus.

Methodology:

Line 154: please add a reference related to the PCR methodology reported.

Line 183: please report the year of edition of the CLSI guidelines used.

Tables (except for tables reported in supplementary materials) should all be formatted, because they are not clearly understandable.

Round 2

Reviewer 1 Report

Authors have well addressed the reviewer's comcerns. 

Minor editing of English language required.
